

# Boundary states of three dimensional topological order and the deconfined quantum critical point

Wenjie Ji[1,2], Nathanan Tantivasadakarn[3] and Cenke Xu[1]

**1** Department of Physics, University of California, Santa Barbara, CA 93106, USA
**2** Kavli Institute for Theoretical Physics, University of California, Santa Barbara, CA 93106, USA
**3** Walter Burke Institute for Theoretical Physics and Department of Physics, California Institute of Technology, Pasadena, CA 91125, USA

## Abstract

We study the boundary states of the archetypal three dimensional topological order, i.e. the three dimensional $\mathbb{Z}_2$ toric code. There are three distinct elementary types of boundary states that we will consider in this work. In the phase diagram that includes the three elementary boundaries there may exist a multi-critical point, which is captured by the so-called deconfined quantum critical point (DQCP) with an "easy-axis" anisotropy. Moreover, there is an emergent $\mathbb{Z}_{2,d}$ symmetry that swaps two of the boundary types, and it becomes part of the global symmetry of the DQCP. The emergent $\mathbb{Z}_{2,d}$ (where d represents "defect") symmetry on the boundary is originated from a type of surface defect in the bulk. We further find a gapped boundary with a surface topological order that is invariant under the emergent symmetry.

# 1   Introduction

For many physical systems, the most prominent features happen at the boundary rather than the bulk. The well-known examples of such include the topological insulators and topological superconductor, as well as their generalizations, the so-called "symmetry protected topological" (SPT) states [1, 2]. Although various diagnosis based on the bulk wave functions have been developed for these systems [3–6], most physical observables still happen at the boundary, or defects.

Boundaries of long-range entangled topological states have also attracted much attention. [7–19] For example, the two-dimensional toric code topological order has two types of gapped one-dimensional boundaries, which correspond to the boundary type with condensation of anyon $e$ and $m$ respectively. Furthermore, at the interface between the two types of boundaries, there is a localized Majorana zero mode. In the two-dimensional toric code, there is a transformation that exchanges the $e$ and $m$ anyons, and hence swaps the two boundary types. One can also drive the boundary to a $(1+1)D$ Ising critical point that is invariant under the $e-m$ exchange symmetry. [14] The emergent symmetry acts on the boundary precisely in the manner of the Kramers-Wannier self-duality. [20, 21] Similar boundary phenomena are much less studied for higher dimensional topological orders, even for some of the archetypal topological orders. It has only been recently clarified that, on the boundary of the three dimensional $\mathbb{Z}_2$ topological order, there can be three distinct types of gapped phases. The three phases are condensations of topological excitations with distinct properties, one of which involves a "twist" of the condensation, as will be elaborated later. [22]

In this work, we consider the phase diagram including the three types of gapped boundaries. We find that depending on the microscopic physics, there could exist a multi-critical point among the three gapped boundaries, and this multi-critical point is captured by the deconfined quantum critical point proposed in the context of spin-1/2 quantum magnet [23, 24]. Moreover, we demonstrate that there also exists an emergent $\mathbb{Z}_{2,d}$ (0-form) duality symmetry which swaps two of the boundary types, and the action of the $\mathbb{Z}_{2,d}$ symmetry can be implemented by "sweeping" an invertible defect through the system.

The three boundary types mentioned above are representative boundaries without anyons that are only localized at the boundary. If we loosen the last constraint, more exotic gapped boundaries can exist. In particular, we demonstrate that there exists a gapped boundary which preserves the emergent $\mathbb{Z}_{2,d}$ symmetry. This boundary can alternatively be constructed by *gauging* a bulk 3$D$ SPT state with $\mathbb{Z}_{2,d} \times \mathbb{Z}_2$ symmetry with a boundary that has topological order.

## 2 Three classes of gapped boundaries

### 2.1 Constructions

— *Gauging a short-range-entangled state*

The three distinct elementary types of boundaries can be constructed explicitly through two procedures. The first construction is to *gauge* the $\mathbb{Z}_2$ symmetry in a three-dimensional short-range entangled state with a gapped boundary. This procedure turns the system into a long-range entangled topological order. In the $3D$ bulk, we will always start with a trivial disordered state (a paramagnet) symmetric under the $\mathbb{Z}_2$ symmetry. On the $2D$ boundary, however, we can consider three different types of short-range entangled states at the $2D$ boundary: (1) The ordered state that spontaneously breaks the $\mathbb{Z}_2$ symmetry; (2) the trivial disordered product state or a $\mathbb{Z}_2$ paramagnet; (3) the $\mathbb{Z}_2$ non-trivial symmetry protected topological state (often referred to as the "Levin-Gu" state [25]).

After gauging the $\mathbb{Z}_2$ symmetry, the bulk will become a $\mathbb{Z}_2$ topological order, and the three different short-range entangled $2D$ boundary states will turn into the following three gapped boundaries:[1]

(1) The "Higgsed" boundary, which is the descendant of the $\mathbb{Z}_2$ ordered short-range entangled state after gauging. At the Higgsed boundary, the point-like $e$ anyon in the bulk condenses at the boundary, which freezes the dynamics of the $\mathbb{Z}_2$ gauge field at the boundary through the Higgs mechanism. The $m-$loop excitation in the bulk cannot terminate at the Higgsed boundary as it braids nontrivially with the condensed $e-$anyon. This scenario is the analogue of the $e-$boundary of the $2D$ toric code phase.

(2) The "deconfined" boundary, which is obtained by gauging the $3D$ bulk with a trivial disordered $\mathbb{Z}_2$ paramagnet at the boundary. This boundary condition allows the $m-$loop to terminate at the boundary, hence we can also view this phase as the analogue of the $1d$ $m-$boundary of the $2D$ toric code. The $e-$anyon from the bulk, and the $m-$loop termination at the boundary will have mutual semionic statistics at the boundary. Nevertheless, the end points of the $m-$loop which terminate on the boundary always costs energy proportional to the length of the string. Thus, there is a string tension between two terminations attached with a single $m-$loop.

(3) The "twisted" boundary is obtained by gauging the $3D$ bulk with the Levin-Gu SPT at the boundary. It is known that if we gauge the two dimensional Levin-Gu state the resulting state is the double-semion topological order. Namely, the $\mathbb{Z}_2$ flux gets transmuted to have semionic statistics instead of bosonic statistics of the toric code. Similarly, on the twisted boundary, the $m-$string in the bulk can also terminate at this boundary like the deconfined boundary. However, the end point of the $m$-string has semionic statistics.

— *Layered construction*

The other construction that also gives rise to the above gapped boundaries is the so-called "layered construction" [22, 26, 27], which consists of starting with stacks of $\mathbb{Z}_2$ toric codes, and performing condensation of composite excitations between layers. The 3D toric code can then be obtained by condensing pairs of $e$ excitations from adjacent layers, causing the $e$ anyon to become mobile in the third direction. On the other hand, the $m$ anyons in each layer are confined, but a product of $m$ anyons braids trivially with the condensate and therefore forms the $m$ loop of the toric code.

---

[1]Abstractly, these topological boundaries are characterized by the so-called Lagrangian algebras of the higher category that describes the topological order. Even though the development of the theory is still on-going, simple examples such as that for the $\mathbb{Z}_2$ topological order have been worked out.

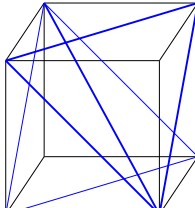 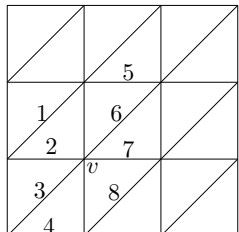

Figure 1: (left) A cube of the three dimensional lattice is triangulated into six tetrahedrons. (right) The tilted triangular lattice on the boundary.

The condensation near the top-most layer of the toric code determines which gapped boundary is obtained as follows

(1) The Higgsed boundary is obtained by also condensing $e$ in the top-most layer. This corresponds to condensing the $e$ particle of the 3D toric code on the boundary.

(2) The deconfined boundary is obtained by condensing only $e$ pairs in adjacent layers, analogous to the condensation in the bulk. This allows the $m$ loop to terminate on the boundary

(3) The twisted boundary is obtained by replacing the 2$D$ toric code model on the top layer with the double semion (DS) model. Then, we still let the pair of $e-$anyons from every two adjacent layers condense. In particular, on the top two layers, the pair to condense is the bound state of the boson, usually named $s\bar{s}$, in the DS layer and the $e$ particle in the second (toric code) layer. This results in the $m$-loop of the 3D toric code terminating as a semion $s$. And the boson $s\bar{s}$ can freely move into the bulk and become the $\mathbb{Z}_2$ charge.

## 2.2 Exactly solvable limits

Let us give the microscopic models in exactly solvable limits corresponding to the gapped boundaries, also called "topological boundaries" [22,28]. Our system is on a three dimensional lattice. The lattice is a triangulation of the cubic lattice, each cube is triangulated into six tetrahedrons, as shown on the left of Fig. 1. We consider a smooth termination of the three dimensional lattice, so that the boundary is on a two-dimensional tilted triangular lattice, as illustrated in Fig. 1.

The bulk Hamiltonian is given by

$$H^{\text{bulk}} = -\sum_v A_v - \sum_\Delta B_\Delta \,, \qquad A_v = \prod_{l \ni v} X_l \,, \qquad B_\Delta = \prod_{l \in \Delta} Z_l \,, \qquad (1)$$

such that all the 14 edges in $A_v$, as well as all the 3 edges of a triangle $\Delta$ in $B_\Delta$ are within the three dimensional lattice with a boundary. Then any state that is the ground state of $H^{\text{bulk}}$ is a boundary state. The wavefunctions of these states are only distinct near the boundary. More precisely, one can show that for any two boundary states, their reduced density matrices in any bulk region, say a region within order 1 distance from the boundary, are identical, by virtue of the topological order.

In the boundary Hamiltonian $H^{\text{bdry}}$, we allow any local terms that commute with all terms in $H^{\text{bulk}}$. What terms are there? The boundary Hilbert space is not a local tensor product Hilbert space, as there are constraints if we restrict ourselves on the ground state subspace of the bulk Hamiltonian. First, note that for any triangle $\Delta$ on the boundary $\mathcal{B}$ (see the right in Fig. 1), there is a constraint

$$B_{\Delta \in \mathcal{B}} = 1 \,, \qquad \prod_{l \in \mathcal{C}^{\text{bdry}}} Z_l = 1 \,, \qquad (2)$$

where $\mathcal{C}^{\text{bdry}}$ is any non-contractible loop on the boundary. The second constraint is the following,

$$G \equiv \prod_{\text{bulk } v} A_v = \prod_{l \ni \text{boundary } v,\, l \in \text{bulk}} X_l = 1, \tag{3}$$

as this operator is a product of bulk star terms.

Given these two bulk constraints, we can find three fixed-point boundary Hamiltonians: Each Hamiltonian has local terms that commute with each other, and is translation invariant. [22]

(1) The "Higgsed" boundary. On this boundary, the $e$ particles are condensed, as well as the Cheshire charge - a string where $e$ particles are condensed. The fixed point Hamiltonian is

$$H^e = -\sum_{\text{boundary } l} Z_l. \tag{4}$$

(2) The "deconfined" boundary. On this boundary, the $m$-loop is condensed. The $m$-loop excitation in the bulk can open up, and terminate on the boundary. The fixed point Hamiltonian is

$$H^m = -\sum_{\text{boundary } v} A_v, \qquad A_v = \prod_{l \ni v} X_l, \tag{5}$$

where each star term $A_v$ is a product of ten Pauli-$X$ operators.

(3) The "twisted deconfined" boundary. Here, the twisted $m$-loop is condensed. The fixed point Hamiltonian is

$$H^{\text{twisted } m} = -\sum_{\text{boundary } v} A_v^{\text{twisted}}, \tag{6}$$

$$A_v^{\text{twisted}} = A_v \omega(g_1, g_2 g, g)\omega(g_2 g, g, g_3)\omega(g, g_3, g_4)\omega(g_5, g_6 g, g)\omega(g_6 g, g, g_7)\omega(g, g_7, g_8),$$

where $g$ is the generator of the $\mathbb{Z}_2$ group, $\omega(g_i, g_j, g_k)$ is a representative of $[\omega]$, the non-trivial class in $H^3[\mathbb{Z}_2, U(1)]$. Here, our convention is that the only non-trivial element is $\omega(g, g, g) = -1$. And $g_i, i = 1, \cdots 8$ label the states on the 8 edges around a vertex $v$ shown on the right of Fig. 1.

One can see that, up to additional Pauli $X$ operators on edges pointing into the bulk, the terms in $H^m$ are the "star terms" in the two-dimensional toric code model, and the terms in $H^{\text{twisted } m}$ are equivalent to the "star terms" in the two-dimensional double semion model [29], up to a finite-depth quantum circuit on the boundary. Nevertheless, since the "plaquette terms" $B_\Delta = 1$ for $\Delta$ on the boundary, are part of bulk Hamiltonian (1). Thus, the magnetic quasi-particle excitation is not present when the bulk is in the ground state. Only a single type of point-like excitations appears on the two types of deconfined boundaries, which is the $\mathbb{Z}_2$ electric charge.

The fixed-point wavefunctions of the toric code model with the three types of boundaries are condensations of surfaces, as illustrated in Fig. 2 and Fig. 3. More precisely, on the background that $Z_l = 1$ on all links, a closed surface $\partial \mathcal{V}$ in the bulk is created by $\prod_{v \in \mathcal{V}} A_v$; an open surface $\partial \mathcal{V}$ that lands on the boundary is also created by $\prod_{v \in \mathcal{V}} A_v$, except that for those vertices $v$ on the boundary, $A_v$ is given in Eq. (5). For the twisted boundary, the sign of the wavefunction amplitude counts the parity of the number of open surfaces on the boundary. The wavefunctions are obtained from the fixed-point wavefunctions of the 3D short-range entangled states with three types of short-range entangled boundaries via gauging. In particular, the basis state $|\{g_i\}\rangle$, with $g_i \in \{0, 1\}$ on the site $i$, is mapped to $|\{g_{ij} = (g_i + g_j) \mod 2\}\rangle$.

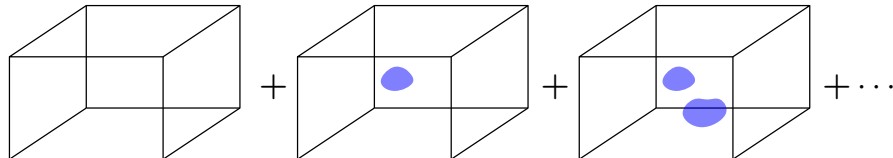

Figure 2: The ground state wavefunction with the "Higgsed" boundary. The wavefunction is a superposition of configurations with closed surfaces.

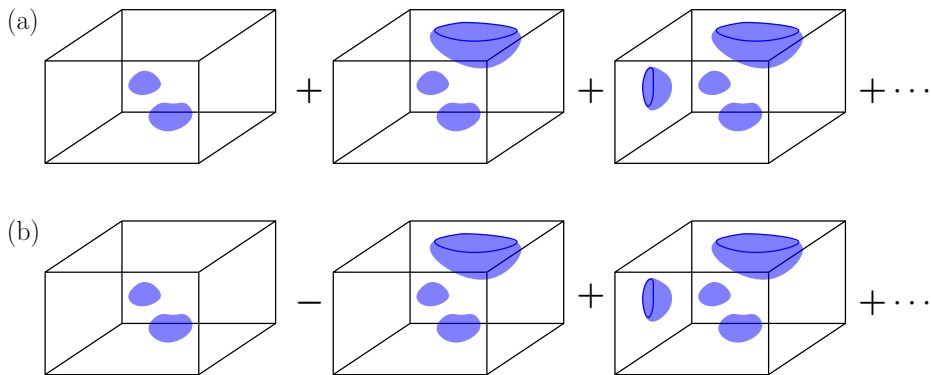

Figure 3: The ground state wavefunctions with (a) the "deconfined" boundary, and (b) the "twisted" deconfined boundary. In both cases, the wavefunctions are superpositions of closed surfaces, as well as open surfaces that terminate on the boundary. The difference comes from the relative phase between the two wavefunctions for those configurations with odd number of open surfaces.

## 2.3 More gapped boundaries with surface anyons

The above gapped boundaries can host surface excitations which can freely move into the bulk. However, there can be further boundaries where some surface excitations *cannot* move freely into the bulk. One trivial way to achieve this is to simply stack an arbitrary $(2+1)D$ topological order onto the boundary.

Here, we demonstrate that there are further boundaries that are beyond stacking with $2D$ topological orders. The construction is very similar to that of the twisted boundary: we replace the top-most toric code layer with a different topological order, and condense certain bosonic anyons. Consider the $3D$ toric code with the deconfined boundary. The bulk excitations are generated by an order-two particle $e_{3D}$ and the loop excitation $m_{3D}$. Now, consider stacking on top a two-dimensional $\mathbb{Z}_4$ toric code which has order-four anyons generated by $e_{2D}$ and $m_{2D}$ where $e_{2D}^4 = m_{2D}^4 = 1$. If we now condense the pair $e_{2D}^2 e_{3D}$ near the top, the anyon $e_{2D}^2$ can now move into the bulk, since it is identified with $e_{3D}$. Nevertheless, $e_{2D}$ is still deconfined only on the boundary.

For the flux sector, originally $m_{3D}$ is allowed to terminate on the boundary. However, since $e_{2D}^2 e_{3D}$ is condensed, this forces the end point of $m_{3D}$ to be bound with $m_{2D}$ in order to braid trivially with the condensate. As a result, the point particles on the boundary are reduced to the set $\{1, e_{2D}, e_{2D}^2, e_{2D}^3\} \times \{1, m_{2D}^2\}$, where $e_{2D}^2$ can move into the bulk. We illustrate this construction in Fig. 4.

Let us provide an explicit lattice realization of this boundary. We replace the qubits on the boundary of the 3D $\mathbb{Z}_2$ toric code with $\mathbb{Z}_4$ qudits and denote the generalized Pauli operators which satisfy $\mathcal{Z}\mathcal{X} = i\mathcal{X}\mathcal{Z}$. Because we are dealing with $\mathbb{Z}_4$ variables on the boundary, we need to pick an orientation of each edge $O_e = \pm 1$. The bulk stabilizers are as in Eq. (1), while the

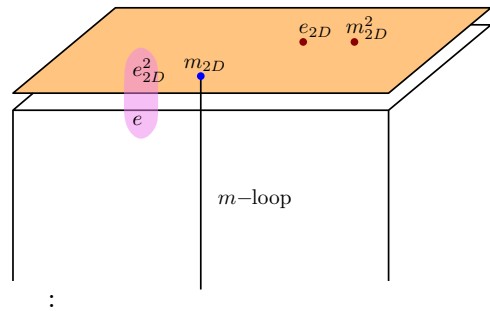

Figure 4: The $3D$ $\mathbb{Z}_2$ topological order with a surface topological order, via stacking a surface topological order and surface condensation. Here, $e_{2D}$ and $m_{2D}^2$ from $\mathbb{Z}_4$ toric code generates surface anyons, among which $e_{2D}^2$ can move freely into the bulk. And $m_{2D}^2$ only exists at the termination of $m-$loop.

vertex stabilizers on the boundary are

$$A_v^{\text{bdry}} = \prod_{l \supset v; l \in \partial M} \mathcal{X}_l^{O_l} \prod_{l \supset v; l \in M} X_l \,. \tag{7}$$

As for the boundary plaquettes, we define two types of plaquettes. If the plaquette has links that are completely in the boundary then we define

$$B_{p_{\parallel}}^{\text{bdry}} = \prod_{l \subset p; l \in \partial M} \mathcal{Z}_l^{O_l} \,. \tag{8}$$

On the other hand, if the plaquette contains links pointing into the bulk, we define

$$B_{p_{\perp}}^{\text{bdry}} = \prod_{l \subset p; l \in \partial M} \mathcal{Z}_l^2 \prod_{l \subset p; l \in M} Z_l \,. \tag{9}$$

As an explicit example, for a cubic lattice, the boundary stabilizers are

$$A_v^{\text{bdry}} = \begin{array}{c} \mathcal{X}^{\dagger} \\ \mathcal{X} \underset{\mathcal{X}}{\diagup} \mathcal{X}^{\ddagger} \\ X \end{array} \tag{10}$$

$$B_{p_{\parallel}}^{\text{bdry}} = \begin{array}{c} \mathcal{Z} \\ \mathcal{Z} \underset{\mathcal{Z}^{\dagger}}{\diagup} \mathcal{Z}^{\dagger} \end{array} \,, \qquad B_{p_{\perp}}^{\text{bdry}} = \begin{array}{c} \mathcal{Z}^2 \\ Z \underset{\phantom{Z}}{\diagup} \\ Z \underset{Z}{\diagup} \end{array} \,, \begin{array}{c} \mathcal{Z}^2 \\ Z \underset{\phantom{Z}}{\mid} Z \\ Z \end{array} \tag{11}$$

Let us now check the boundary excitations. The stabilizer $A_v^{\text{bdry}}$ can be violated by a string of $\mathcal{Z}$ on the direct lattice on the boundary. The eigenvalues $\pm i$ corresponds to the anyon $e_{2D}$ and $e_{2D}^3$ on the boundary. These anyons are deconfined only on the boundary: they cannot move into the bulk. On the other hand, the anyon $e_{2D}^2$ corresponding to $A_v^{\text{bdry}} = -1$ can move into the bulk using $Z_l$ for a link $l$ which ends on the boundary vertex $v$.

Next we analyze the flux excitations. The loop $m_{3D}$ in the bulk can be excited with a membrane of $X_l$ on the dual lattice as usual. However, when the loop ends on the boundary, we must act with $\mathcal{X}$ so that $B_{p_{\perp}}^{\text{bdry}}$ is not excited. This causes $B_{p_{\parallel}}^{\text{bdry}}$ to be excited with eigenvalue $\pm i$. This corresponds to the fact that the end point of the loop in the bulk is attached with $m_{2D}$ of the $\mathbb{Z}_4$ toric code. Finally, the point particle $m_{2D}^2$ can be created with $\mathcal{Z}^2$ on the dual lattice on the boundary.

We can also relate the above boundary to a $\mathbb{Z}_2$ symmetric state. That is, by ungauging both the bulk and boundary, the bulk becomes a $\mathbb{Z}_2$ symmetric paramagnet, while the boundary realizes a $\mathbb{Z}_2$ symmetry-enriched $\mathbb{Z}_2$ toric code, where the $e$ anyon carries fractionalized charge of the $\mathbb{Z}_2$ symmetry.

We remark that a more general family of such boundaries can be constructed by stacking on top of the "deconfined" boundary a general anyon theory described by a modular tensor category $\mathcal{C}_{2D}$ that contains a $\mathbb{Z}_2$ boson $b_{2D}$ and condensing the pair $b_{2D}e_{3D}$ near the top. The resulting anyons on the boundary then can be described by a premodular subcategory containing all anyons that braid trivially with $b_{2D}$ (since all the anyons that braid are now attached to the end points of the flux loops). To obtain the above boundary by gauging, we start with a $\mathbb{Z}_2$ paramagnet in the bulk, and place a $\mathbb{Z}_2$ SET on the boundary. This SET corresponds to the $G$-crossed braided fusion category that results from condensing $b_{2D}$ with $G = \mathbb{Z}_2$.

Let us further remark that from the point of view of gauging, we can also embed such 2+1D SETs in the 3+1D bulk instead of on the boundary. Upon gauging, this gives rise to a codimension one non-invertible defect of the toric code. Such non-invertible defects have an immediate relation to defect network constructions of fracton and hybrid fracton orders [30–34]. In particular, starting with a "foliation" of 2+1D SETs and gauging the $\mathbb{Z}_2$ symmetry in the bulk reproduces exactly the 1-foliated hybrid fracton models.

## 2.4 Effective field theories

The three elementary types of boundaries of the 3D $\mathbb{Z}_2$ topological order can also be described by effective field theories. Our starting point is a generalization of the Luttinger liquid theory that describes the boundary of two-dimensional Abelian topological orders, which we reviewed in Appendix A. A family of Abelian topological orders in three spatial dimensions can be described by a topological action of the $BF$ kind:

$$\mathcal{S}^{\text{bulk}} = \frac{N}{2\pi} \int_{\mathcal{M}} B \wedge dA, \tag{12}$$

where $A$ and $B$ are the dynamical one-form and two-form fields, respectively, $\mathcal{M}$ is the four-dimensional spacetime, and $N \in \mathbb{Z}$ denotes the level. The physical meaning of the two-form field $B$ is that it is the dual of the current of the charge-$N$ matter field that couples to the one-form gauge field $A$: $J \sim dB$. The charge-$N$ matter field condenses and Higgses the one-form gauge field $A$ to a $\mathbb{Z}_N$ gauge field, i.e. the bulk becomes a $\mathbb{Z}_N$ topological order. If the spacetime manifold $\mathcal{M}$ is closed, the action is invariant under the gauge transformation of $A \to A + d\varphi$ for a zero-form field $\varphi$ and $B \to B + d\chi$, for a one-form field $\chi$.

To describe the boundary, we may begin with the following action:

$$\mathcal{S}_0 = \frac{1}{2\pi} \int dt\, dx\, dy \left[ N \partial_t \phi \left( \partial_x b_y - \partial_y b_x \right) + V[\phi, b] \right], \tag{13}$$

where $\phi$ is a zero-form field, $b$ is a one-form field, and $V[\phi, b]$ are some non-universal terms. There are two types of quasi-excitations on the boundary: the point-particle excitation created by $e^{i\phi}$, and loop excitation $e^{i\oint_C b}$, and they are coupled to the gauge field $A$ and $B$ respectively. Local excitations, which can be created by microscopic degrees of freedom locally, are at least composites of these quasi-excitations and have trivial statistics.

Now we consider the case with $N = 2$, the $BF$ theory in the bulk describes the $\mathbb{Z}_2$ topological order. Due to the fusion rule of the quasi-particle $e \times e = 1$, and that of the quasi-loop $m \times m = 1,$[2] we know that there are at least two local excitations $e^{i2\phi}$ and $e^{i2\oint_C b}$. A more

---

[2]We abuse the notation and use $m$ to label the $\mathbb{Z}_2$ flux loop excitation in three-dimensional topological order as well.

generic action for the boundary should include the following local terms:

$$\delta \mathcal{S} = -\int dt\,dx\,dy \left( u_1 \cos 2\phi + \sum_{\mathcal{C}} u_{\mathcal{C}} \cos 2 \oint_{\mathcal{C}} b \right), \tag{14}$$

where $\sum_{\mathcal{C}}$ is a formal sum of all possible closed loop configurations. In the ground state, when the quasi-particle condenses on the boundary, $\langle e^{i\phi} \rangle \neq 0$, the gauge field $A$ is "frozen" due to the Higgs mechanism at the boundary. The condensation of the quasi-particle at the boundary can be caused by the Hamiltonian Eq. 4. In the Higgsed phase of $A$, $A = d\phi$, and the boundary action reduces to

$$\mathcal{S}_e^{\text{bdry}} = \frac{2}{2\pi} \int_{\partial \mathcal{M}} \phi \wedge dB. \tag{15}$$

This action, combined with the bulk action, is invariant under the zero-form gauge transformation $A \to A + d\varphi$, $\phi \to \phi - \varphi$, and the one-form gauge transformation $B \to B + d\chi$.

In the opposite limit, i.e. the "deconfined boundary", the boundary Hamiltonian Eq. 5 can cause the loop excitation $e^{i\oint_{\mathcal{C}} b}$ to condense at the boundary, for loops $\mathcal{C}$ parallel with the boundary. In the condensate of the loop excitations, the two-form field $B$ is identified with $db$. Now the gapped boundary is described by the topological action,

$$\mathcal{S}_m^{\text{bdry}} = \frac{2}{2\pi} \int_{\partial \mathcal{M}} b \wedge dA. \tag{16}$$

This action, combined with the bulk action, is invariant under the zero-form gauge transformation $A \to A + d\varphi$, and the one-form gauge transformation $B \to B + d\chi$, $b \to b - \chi$.

There is also a third choice of the boundary topological action:

$$\mathcal{S}_{\text{twisted } m}^{\text{bdry}} = \frac{1}{2\pi} \int_{\partial \mathcal{M}} (2b \wedge dA - A \wedge dA). \tag{17}$$

This action, combined with the bulk action, is also invariant under the zero-form gauge transformation $A \to A + d\varphi$, and the one-form gauge transformation $B \to B + d\chi$, $b \to b - \chi$. In fact, $\mathcal{S}^{\text{bdry}} = \frac{1}{2\pi} \int_{\partial \mathcal{M}} (2b \wedge dA - kAdA)$ are allowed choices with integer $k$ for a bosonic system; nevertheless when $k = 2k'$ with integer $k'$, the action is equivalent to $\mathcal{S}_m^{\text{bdry}}$ up to a field redefinition $b \to b - k'A$. Hence the only physically distinct boundary action is the one with $k = 1$.

Let us analyze the excitations on the boundary. We start with the second boundary condition described by

$$\mathcal{S}_m^{\text{bdry}} = \frac{2}{4\pi} \int_{\partial \mathcal{M}} (b \wedge dA + A \wedge db). \tag{18}$$

Even though $b$ comes from the pure gauge in the bulk $B = db$, it is a dynamical $U(1)$ gauge field on the boundary. Then comparing Eq (18) with Eq (A.1), we recognize that this action is the two-component Chern-Simons action with a $K$ matrix which is the same as that of the 2D $\mathbb{Z}_2$ topological order.

Similarly, on the third type of boundary, the action can be rewritten as

$$\mathcal{S}_{\text{twisted } m}^{\text{bdry}} = \frac{1}{4\pi} \int_{\partial \mathcal{M}} (2b \wedge dA + 2A \wedge db - 2A \wedge dA). \tag{19}$$

This is the topological action describing the twisted $\mathbb{Z}_2$ topological order. With this twisted action, the point gauge charge of $A$ remains a deconfined bosonic excitation on the boundary. Now let us discuss the terminations of $m-$loops on the boundary. Consider an excited state

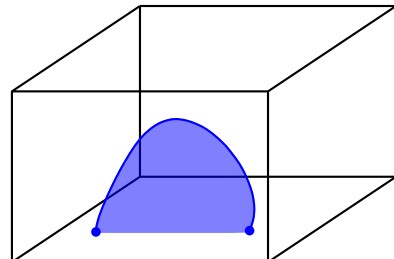

Figure 5: A $\mathbb{Z}_2$ flux string with two end-points terminating on the boundary.

in the topological order, where an open $m$-string ends on the boundary at two points. Such excitation only exists when the $m$-loop is condensed on the boundary, described either by $\mathcal{S}_m^{\text{bdry}}$ or $\mathcal{S}_3^{\text{bdry}}$. The statistics of the end-point of the $m-$loop corresponds to the charge of $b$, i.e. the charge vector $\mathbf{m} = (0,1)^T$ of the deconfined boundary theory Eq. (18) or the twisted boundary given by Eq. (19), where $K = \begin{pmatrix} 0 & 2 \\ 2 & 0 \end{pmatrix}$ or $K = \begin{pmatrix} -2 & 2 \\ 2 & 0 \end{pmatrix}$ respectively. Its self-statistics is given by $\theta_m = \pi \mathbf{m}^T K^{-1} \mathbf{m}$. On the deconfined boundary, $\theta_m = 0$; while on the twisted deconfined boundary, $\theta_m = \frac{\pi}{2}$, i.e. the end-points of the $m-$loops are self-semionic.

## 3 Emergent symmetry and its defect

There is a unitary operator which leaves the ground state subspace of the toric code on a closed manifold invariant, and thus can be taken as generating an emergent (non-onsite) 0-form $\mathbb{Z}_2$ symmetry in the ground state subspace.

Nevertheless, if the manifold has a boundary, this unitary operator exchanges the "deconfined" and "twisted" boundaries. [35,36] The unitary, if acting only on a region $\mathcal{V}$ in the bulk, creates a codimension-1 invertible symmetry defect. Physically speaking, there is a "gauged Levin-Gu state" on the defect.[3] More precisely, the defect is obtained by first embedding a codimension-1 defect containing the Levin-Gu SPT state into a three-dimensional $\mathbb{Z}_2$ paramagnet. Then, after gauging the $\mathbb{Z}_2$ symmetry, the codimension-1 defect becomes the symmetry defect of this emergent $\mathbb{Z}_2$ symmetry of the toric code.

Explicitly, the unitary operator acting on a region $\mathcal{V}$ is,

$$U(\mathcal{V}) = \prod_{\triangle_{0123} \in \mathcal{V}} e^{i\frac{\pi}{8}\left(1 + \sum_{i \neq j}(-1)^{i+j} Z_{ij} + Z_{01} Z_{23}\right)}, \tag{20}$$

where $0,1,2,3$ labels the four vertices of a tetrahedron, or rather a branched 3-simplex [36]. The formula comes from gauging the symmetric finite depth circuit that create a two-dimensional $\mathbb{Z}_2$ SPT on the boundary $\partial \mathcal{V}$ of a volumn $\mathcal{V}$, which we show in the Appendix B (see also, Ref. [37] for a similar unitary defined on the cubic lattice).

If we begin with the $\mathbb{Z}_2$ topological order with a "deconfined" boundary, and take $\mathcal{V}$ to be the whole system, then the (non-onsite) unitary $U$ commutes with $H^{\text{bulk}}$ in the subspace where $B_\triangle = 1$, yet changes the "deconfined" to the "twisted" boundary. That is, up to plaquette terms $B_\triangle$,

$$U A_{\text{boundary } v} U^{-1} = A_{\text{boundary } v}^{\text{twisted}}. \tag{21}$$

---

[3]In a purely two-dimensional system, a gauged Levin-Gu state is a non-invertible state with double semion topological order. However, here the defect created by the finite-depth circuit is invertible. There is no deconfined point excitation localized on the defect, as described in Fig.6.

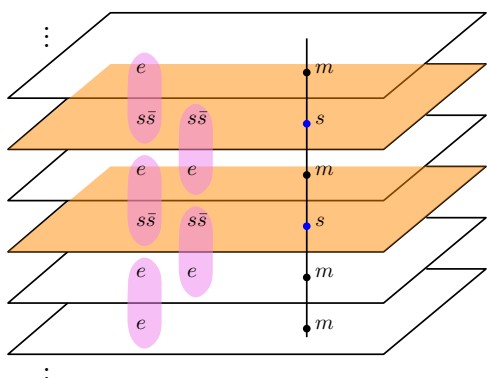

Figure 6: The $\mathbb{Z}_2$ flux loop piecing through the invertible codimension-1 defect (orange surface). On the defect, semion $s$ is attached to the $m$-loop, the boson $s\bar{s}$ can move into the bulk as $\mathbb{Z}_2$ gauge charge.

The unitary operator $U$ can be considered as a finite-depth quantum circuit on a three-dimensional qubit system that swaps the two gapped boundaries.

## 3.1 Self-statistics of the termination of the $m-$loop

The symmetry defect $U(\mathcal{V})$ also changes the statistics of the end-points of the $m$-loop. One way to think of this is through a layered construction, as illustrated in Fig. 6. The three dimensional $\mathbb{Z}_2$ gauge theory can be prepared with layers of toric code, followed with condensing the pair of $\mathbb{Z}_2$ charges from neighboring layers. We can create the domain wall defects on two layers (orange layers in the figure) by acting with the unitary in the region between the two layers. This topological state with defects can also be prepared with a layered construction. Here, we replace two layers of toric code models with double semion models, and still condense the pair of bosonic quasi-particles from neighboring layers. In this state, the deconfined excitations are the mobile $\mathbb{Z}_2$ charge, as well as the $\mathbb{Z}_2$ flux. The $\mathbb{Z}_2$ flux is now a string of magnetic quasi-particles in all layers. In particular, in the defected layers, the magnetic quasi-particles are semions.

Because of the condensed bosonic pairs, the even number of semions on the flux string are identical to the even number of anti-semions, through fusing with the condensed pairs. See Fig. 7 for the illustration.

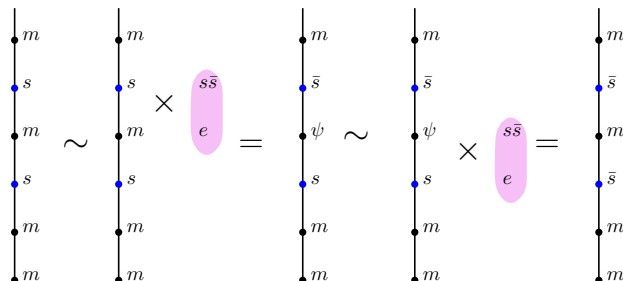

Figure 7: A pair of intersection points of the flux string and invertible codimension-1 defect are either all self-semionic or all self-anti-semionic.

Table 1: Excitations of the deconfined boundary with an additional chiral semion topological order $s'$. The first four excitations are deconfined point excitations on the boundary, the first two can move into the bulk; while the last four excitations are confined as they are attached to flux loops in the bulk. Under the action of $\mathbb{Z}_{2,d}$, which adds the term $\frac{2}{4\pi}AdA$ to the boundary, the end point of the loop which is a boson $m$ is transmuted into to a semion $s$. The excitations after the transmutation can be identified by an isomorphism $W$ to the original excitations, showing that the $\mathbb{Z}_{2,d}$ symmetry action swaps $s'$ and $s'e$ on the boundary, and transmutes the self-statistics of the flux loop end points by a phase $i$.

| Vector | excitation | statistics |  | excitation | statistics |  | excitation |
|---|---|---|---|---|---|---|---|
| $(0,0,0)$ | $1$ | $1$ |  | $1$ | $1$ |  | $1$ |
| $(1,0,0)$ | $e$ | $1$ |  | $e$ | $1$ |  | $e$ |
| $(0,0,1)$ | $s'$ | $i$ |  | $s'$ | $i$ |  | $s'e$ |
| $(1,0,1)$ | $s'e$ | $i$ | $\xrightarrow{\mathbb{Z}_2^d}$ | $s'e$ | $i$ | $\underset{W}{\simeq}$ | $s'$ |
| $(0,1,0)$ | $m$ | $1$ |  | $s$ | $i$ |  | $ms'$ |
| $(0,1,1)$ | $ms'$ | $i$ |  | $ss'$ | $-1$ |  | $f$ |
| $(1,1,0)$ | $f=me$ | $-1$ |  | $\bar{s}=se$ | $-i$ |  | $fs'$ |
| $(1,1,1)$ | $fs'$ | $-i$ |  | $\bar{s}s'$ | $1$ |  | $m$ |

## 3.2 A "deconfined" gapped boundary symmetric under the emergent symmetry

If we consider the action of the invertible defect as an additional $\mathbb{Z}_{2,d}$ symmetry at the boundary, which can be implemented by "sweeping" the invertible defect through the entire bulk, then only the Higgs boundary is invariant under this $\mathbb{Z}_{2,d}$ symmetry. However, it is possible to have a boundary that preserves the $\mathbb{Z}_{2,d}$ symmetry and also have deconfined point excitations.

We first recall that the "deconfined" boundary is transformed into the "twisted" boundary under the $\mathbb{Z}_{2,d}$ symmetry. Specifically, the statistics of the $m$-loop at the end point is transmuted from a boson to a semion. Let us now show that the "deconfined" boundary and the "twisted" boundary become equivalent by adding a 2+1D chiral semion topological order to the boundary. Thus, by stacking the chiral semion onto either boundaries, we obtain a boundary with surface topological order symmetric under the emergent symmetry.

Let us consider the following topological action on the boundary,

$$S_{s'}^{\text{bdry}} = \frac{1}{2\pi} \int_{\partial M} (2b \wedge dA + c \wedge dc) . \tag{22}$$

This theory describes a stack of chiral semion topological order, which is given by the $U(1)$ Chern-Simons theory at level two associated with $c$, a dynamical $U(1)$ field, onto the "deconfined" boundary given in (16). The K-matrix of this boundary can be written as

$$K_{s'} = \begin{pmatrix} 0 & 2 & 0 \\ 2 & 0 & 0 \\ 0 & 0 & 2 \end{pmatrix}, \tag{23}$$

where we choose the basis such that the $e$ anyon, the end point of the loop excitation $m$ and the newly introduced semion $s'$ correspond to unit charge vectors $(1,0,0)^T$, $(0,1,0)^T$, $(0,0,1)^T$, respectively. The full list of excitations is summarized in Table 1.

Under the action of the $\mathbb{Z}_{2,\mathrm{d}}$ symmetry, the boundary is transformed into

$$\mathcal{S}_{s'}^{\mathrm{bdry}} \to \frac{1}{2\pi} \int_{\partial M} (2b \wedge dA - A \wedge dA + c \wedge dc)\,. \tag{24}$$

That is,

$$K_{s'} \to \widetilde{K}_{s'} = \begin{pmatrix} -2 & 2 & 0 \\ 2 & 0 & 0 \\ 0 & 0 & 2 \end{pmatrix}\,. \tag{25}$$

As a result, the end point of the loop excitation $m$ (which is originally a boson) gets transmuted to a semion $s$.

However, the two boundaries related by the symmetry are actually equivalent. We can see this by identifying the generating excitations as follows

$$e \simeq e\,, \qquad s \simeq ms'\,, \qquad s' \simeq s'e\,. \tag{26}$$

The above identification preserves the statistics and braiding, and can be encoded as the following transformation on the $K$-matrix

$$W = \begin{pmatrix} 1 & 0 & 1 \\ 0 & 1 & 0 \\ 0 & -1 & 1 \end{pmatrix}\,, \tag{27}$$

where we can now explicitly confirm that

$$W \widetilde{K}_{s'} W^T = K_{s'}\,. \tag{28}$$

In conclusion, we have shown that the deconfined and twisted boundaries are equivalent after stacking with a chiral semion topological order. Moreover, under the symmetry action of $\mathbb{Z}_{2,\mathrm{d}}$ on the boundary, the anyons $s'$ and $s'e$ are permuted, and the termination of $m-$loop has self-statistics transmuted by a phase $i$. This shows that the chiral semion topological order can "absorb" the invertible defect which is pushed onto the boundary under the action of $\mathbb{Z}_{2,\mathrm{d}}$. We remark that it is known that the chiral semion topological order can absorb the Levin-Gu SPT [38–41]. Our result can be viewed as the gauged version of the results above.

## 3.3 A twisted bulk topological order

In the $\mathbb{Z}_2$ topological order, we can describe the presence of the invertible defect on the surface $\Omega$ of a volume $\mathcal{V}$ by a topological action,

$$\mathcal{S}[a,b] = \frac{2}{2\pi} \int b \wedge da - \frac{2}{4\pi^2} \int \delta^{\perp}(\mathcal{V}) \wedge a \wedge da\,, \tag{29}$$

where $\delta^{\perp}(\mathcal{V})$ is the delta-form for the volume $\mathcal{V}$, and $a, b$ are dynamical 1-form and 2-form $U(1)$ gauge fields, respectively.

We can further gauge the $\mathbb{Z}_{2,\mathrm{d}}$ symmetry. The system becomes another topologically ordered phase described by a twisted Dijkgraaf-Witten model [42]. The topological action is

$$\mathcal{S}[a,b] = \frac{2}{2\pi} \sum_{I=1,2} \int b^I \wedge da^I - \frac{2}{4\pi^2} \int a^2 \wedge a^1 \wedge da^1\,, \tag{30}$$

where $a^I, b^I$ for $I = 1, 2$ are dynamical 1-form and 2-form $U(1)$ gauge fields, respectively. The associated 4-cocycle $[\omega] \in H^4[\mathbb{Z}_2 \times \mathbb{Z}_2, U(1)]$ can be represented by

$$\omega(g,h,l,m) = e^{\frac{i2\pi}{N^2} \sum_{IJK} M_{IJK} g_I h_J (l_K + m_K - [l_K + m_K])}\,, \tag{31}$$

where $N = 2$, $g, h, l, m \in \mathbb{Z}_2 \times \mathbb{Z}_2$, $I, J, K = 1, 2$ and the only non-vanishing component of $M_{IJK}$ is $M_{211} = 1$.

Alternatively, this topological order can also be obtained from gauging a $\mathbb{Z}_2 \times \mathbb{Z}_2$ SPT phase characterized by the same 4-cocycle. The topological response theory for the latter is described by

$$\mathcal{S}[A] = -\frac{2}{4\pi^2} \int A^2 \wedge A^1 \wedge dA^1 \,, \tag{32}$$

where $A^1$ and $A^2$ are background $U(1)$ gauge fields.

Correspondingly, let us describe what the gapped boundary that respects the $\mathbb{Z}_{2,\mathrm{d}}$ symmetry becomes after the symmetry is gauged. Specifically, we consider the boundary that has a surface chiral semion topological order shown in Subsection 3.2. Since the $\mathbb{Z}_{2,\mathrm{d}}$ symmetry action swaps $s'$ and $s'e$, after gauging the two surface anyons combine into a single non-abelian anyon with self-statistics $i$ and quantum dimension two. Fusing two such "non-abelian semions" can result in four fusion outcomes corresponding to the sign representations of $\mathbb{Z}_2 \times \mathbb{Z}_{2,\mathrm{d}}$, which are gauge charges that can freely move into the bulk. The deconfined point excitations on this boundary are denoted by the premodular category $\mathrm{Rep}_s(D_4)$. [43, 44].

# 4 Phase transitions between gapped boundaries

## 4.1 Phase diagram of the three boundaries

In the previous section we have discussed three different boundary states of the bulk 3$D$ toric code topological order. More generally, when the bulk is in the ground state, depending on tuning parameters, the boundary system can interpolate between the three phases. For example, the following Hamiltonian describes a two dimensional phase diagram with the three boundaries:

$$H^{\mathrm{bdry}} = -\lambda H^e + (1-\alpha)H^m + \alpha H^{\mathrm{twisted}\ m} + \cdots \,, \tag{33}$$

which includes the weighted sum of the fixed point Hamiltonians. The ellipsis refers to other local terms that can be viewed as perturbations.

A schematic phase diagram of this Hamiltonian is shown in Fig. 8. Along the line $\alpha = 1/2$, the inversion of symmetry defect by $\prod_\triangle U(\triangleright)$ (see Eq. (20)) becomes an extra "emergent" $\mathbb{Z}_{2,\mathrm{d}}$ symmetry of the boundary Hamiltonian. At large $\lambda$ along the line $\alpha = 1/2$, this emergent $\mathbb{Z}_{2,\mathrm{d}}$ symmetry is spontaneously broken, and the system is in either the deconfined or the twisted boundary.

The details near the center of the phase diagram may depend on the microscopic details of the Hamiltonian. [45–48] With proper perturbations to $H^{\mathrm{bdry}}$, a multi-critical point where three phases meet together may be realized at the boundary of our system. A similar phase diagram was studied numerically in Ref. [48] for a 2$D$ system with $\mathbb{Z}_2^3$ symmetry and a similar $\mathbb{Z}_{2,\mathrm{d}}$ symmetry. Note that the only difference from Eq. (33) is that the $\lambda$ perturbation was a next-nearest neighbor Ising interaction, which was chosen so that it respects the full $\mathbb{Z}_2^3$ symmetry. There, it was found that the three phases indeed meet at a multicritical point, which was argued to be in the same universality class as the DQCP. We expect that by explicitly breaking the $\mathbb{Z}_2^3$ symmetry down to the diagonal $\mathbb{Z}_2$ symmetry by a infinitesimal amount of the Hamiltonian $H^e$, the characteristics of the phase diagram remain the same and qualitatively reproduces the phase diagram Eq. (33) without additional terms.

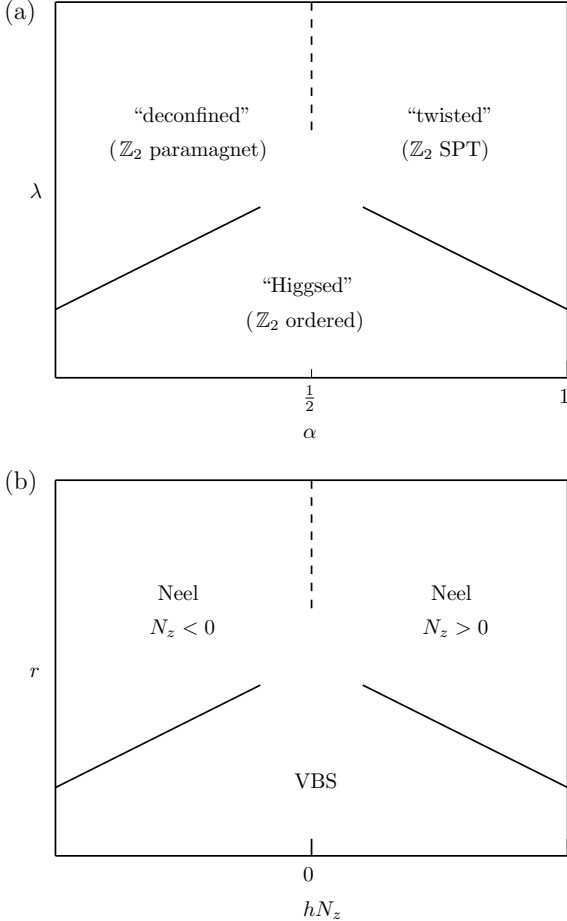

Figure 8: (a) A schematic phase diagram on the boundary of the three-dimensional $\mathbb{Z}_2$ topological order. The topological order with three gapped boundaries can be obtained from "gauging" a three-dimensional $\mathbb{Z}_2$ symmetric short-range entangled state with gapped boundaries (as labeled in parentheses), as we describe in the main text. (b) A schematic phase diagram of quantum magnets described by the noncompact $CP^1$ model with "easy-axis" anisotropy.

## 4.2 Connection to the deconfined quantum critical point

In this subsection, we discuss the connection between the phase diagram Fig. 8 to the deconfined quantum critical point (DQCP) [23, 24]. More precisely, the DQCP corresponds to the multi-critical point in the phase diagram which includes the three short-range entangled 2$D$ states at the boundary of the system before gauging the $\mathbb{Z}_2$ symmetry. Since the $\mathbb{Z}_2$ gauge field does not lead to singular dynamics in the infrared, we expect that the dynamics at the transitions in the phase diagram Fig. 8, especially the potential multi-critical point can still be captured by the DQCP.

The DQCP has a field theory description in terms of the noncompact $CP^1$ (NCCP$^1$) model:

$$\mathcal{L} = \sum_{\alpha=1,2} \left( |D_a z_\alpha|^2 + r|z_\alpha|^2 \right) + g \left( \sum_\alpha |z_\alpha|^2 \right)^2 + u|z_1|^2|z_2|^2 + h \left( \mathbf{z}^\dagger \sigma^z \mathbf{z} \right). \tag{34}$$

There are two main tuning parameters in the system, $r$ and $h$, which should correspond to $\lambda$ and $\alpha$ in the lattice model. Compared with the original DQCP, our system should also have a parameter $u > 0$, which makes the theory an "easy-axis" DQCP in the terminology of quantum magnets. In our case, the photon phase corresponds to the ordered phase that spontaneously

breaks the $\mathbb{Z}_2$ symmetry, which eventually becomes the Higgs phase after the symmetry is gauged.

When $r < 0$, the phase diagram in the DQCP is in the Neel order. The easy axis anisotropy favors either $N^z \sim \mathbf{z}^\dagger \sigma^z \mathbf{z} > 0$ or $N^z < 0$, respectively. Along the line $h = 0$, there is an apparent symmetry in the $\text{CP}^1$ model connecting $N^z > 0$ and $N^z < 0$, which is just a $\mathbb{Z}_2$ symmetry that flips $S^z$ in the quantum magnet context. If all the symmetries of the DQCP are onsite, the theory of DQCP can only be realized on the $2D$ boundary of a $3D$ SPT state [49]; and the $\mathbb{Z}_2$ symmetry that flips $N^z$ is precisely the fermionic duality transformation of the $(2+1)D$ QED with $N_f = 2$ flavors of Dirac fermions [50–52], which is a dual representation of the DQCP [53–55]. This $(2+1)D$ QED can also be used to describe the quantum phase transition between $2D$ bosonic SPT states [56,57], and the fermionic duality transformation changes the level of the bosonic SPT states. Hence this $\mathbb{Z}_2$ symmetry that flips $S^z$ can also be identified as the $\mathbb{Z}_{2,d}$ symmetry discussed previously (Section. 3), i.e. the emergent symmetry that swaps the deconfined and twisted boundaries (which correspond to the $\mathbb{Z}_2$ paramagnet and $\mathbb{Z}_2$ SPT state respectively before gauging). If this $\mathbb{Z}_{2,d}$ symmetry is an exact *onsite* symmetry in our system, the short-range entangled bulk should be a SPT state protected by the $\mathbb{Z}_{2,d} \times \mathbb{Z}_2$ symmetry, whose topological response is given in Eq. (32). The boundary state along the line $h = 0$ either spontaneously breaks the $\mathbb{Z}_2$ symmetry ($r > 0$), or the $\mathbb{Z}_{2,d}$ symmetry ($r < 0$).

When $r < 0$, by tuning $h$, the phase transition between the trivial bosonic insulator and the bosonic SPT phase is expected to be first order due to the easy axis anisotropy. If we fix $h > 0$ (or $h < 0$), the mass degeneracy of the two flavors of the bosonic fields $z_\alpha$ is lifted. By tuning $r$, the transition corresponds to the condensation of one of the two flavors of $z_\alpha$, which is dual to a $3D$ XY transition, if the gauge field $a_\mu$ of the $\text{CP}^1$ theory is noncompact. Since the flux of the gauge field is conserved mod $\mathbb{Z}_2$, the $3D$ XY transition is reduced to a $3D$ Ising transition, for both $h > 0$ and $h < 0$. This is consistent with the common wisdom that the transition between the SPT and the ordered phase with spontaneous symmetry breaking is an ordinary Landau transition.

The DQCP was initially proposed for two dimensional spin-1/2 quantum magnets. The phases involved in the phase diagram of DQCP are all short-range entangled states. But it is also natural for two dimensional spin-1/2 quantum magnets to form topological orders, such as the chiral-semion state, or the $\nu = 1/2$ fractional quantum Hall state if we view the spin as a boson [58]. This spin liquid state can be invariant under the full SU(2) spin symmetry, hence invariant under the $\mathbb{Z}_{2,d}$ symmetry that is a subgroup of the spin rotation. The DQCP is also directly connected to the chiral semion spin liquid. The O(5) nonlinear-Sigma model (NLSM) with a Wess-Zumino-Witten term at level-1 serves as the low energy effective theory of the DQCP on the square lattice [59], and this NLSM can be derived from the so-called "$\pi$−flux" spin liquid state, which is described by a $(2+1)D$ QCD with $N_f = 2$ flavors of Dirac fermion coupled with a SU(2) gauge field. The $\pi$−flux spin liquid state can be driven into the chiral semion state by turning on a mass term of the Dirac fermion spinons.

# 5   Summary and discussions

In this work, we discussed three elementary boundary states of the $3D$ toric code topological order. The three elementary boundary states can be constructed intuitively as gauging the paramagnet boundary, the ordered boundary, and the $\mathbb{Z}_2$ SPT boundary of a trivial symmetric paramagnetic bulk with a global $\mathbb{Z}_2$ symmetry. These boundary states are elementary in the sense that they do not have independent anyons that are localized only at the boundary. Other topological boundary states with anyons restricted at the boundary can be constructed, and one of the topological boundary states is invariant under a $\mathbb{Z}_{2,d}$ symmetry that swaps two of

the elementary boundaries. We also discussed a multi-critical point among the three elementary boundaries, and demonstrated its connection to the deconfined quantum critical point proposed originally in the context of $2D$ quantum magnets.

There is a natural generalization of our results to the boundary states of the $3D$ bulk $\mathbb{Z}_N$ topological order, especially for $N$ being a prime number. For a prime number $N$, there are $N + 1$ types of elementary boundaries, which again can be constructed by gauging the $\mathbb{Z}_N$ paramagnet boundary, the ordered boundary that spontaneously breaks the $\mathbb{Z}_N$ symmetry, and the $N-1$ nontrivial $2D$ SPT states (the $2D$ SPT states with $\mathbb{Z}_N$ symmetry has a $\mathbb{Z}_N$ classification) on the boundary of a trivial symmetric paramagnetic bulk with the $\mathbb{Z}_N$ symmetry.

# Acknowledgments

We thank Chong Wang for interesting discussions on this work. While finishing up this paper, the authors became aware of another independent work which overlaps with ours [60].

**Funding information** W.J. and C.X. are supported by the Simons foundation through the Simons Investigator program. N.T. is supported by the Walter Burke Institute for Theoretical Physics at Caltech.

# A Boundaries of 2D Abelian TO

We begin with an Abelian topological order described by a topological Chern-Simons action,

$$S^{\text{bulk}} = \frac{1}{4\pi} \int_{\mathcal{M}} K_{IJ} a_I \wedge da_J \,, \tag{A.1}$$

where $K$ is a $N \times N$ symmetric integral matrix, whose diagonal elements are even, $a_I$ are one-form fields, and $\mathcal{M}$ is the three-dimensional spacetime manifold. The boundary is a Luttinger liquid system with $U(1)$ symmetries,

$$S_0 = \frac{1}{4\pi} \int dt dx \left( K_{IJ} \partial_x \phi_I \partial_t \phi_J - V_{IJ} \partial_x \phi_I \partial_x \phi_J \right) \,, \tag{A.2}$$

where $\phi_I \in [0, 2\pi)$ are bosonic field, The $U(1)$ symmetry transformations are

$$e^{i\phi_I} \rightarrow e^{i\alpha_I} e^{i\phi_I} \,, \tag{A.3}$$

where $\alpha_I \in [0, 2\pi)$, for $I = 1, \cdots, N$.

Then we turn on backscattering terms of quasiparticles,

$$\delta S = -\int dt dx \sum_{\mathbf{m}} U_{\mathbf{m}} \cos(\mathbf{m}_I \phi_I) \,, \tag{A.4}$$

where $\mathbf{m}$ are integer vectors.

The edge can be gapped due to the backscattering if and only if quasiparticles represented by the integer vectors $\mathcal{M} = \{\mathbf{m}\}$, satisfy the Lagrangian group condition [8, 11]:

1. The quasiparticles in $\mathcal{M}$ are mutually bosonic, $\frac{\theta_{\mathbf{mm}'}}{2\pi} = \mathbf{m}^T K^{-1} \mathbf{m}' \in \mathbb{Z}$ for any $\mathbf{m}, \mathbf{m}' \in \mathcal{M}$.

2. Any quasiparticle, given by an integer vector $\mathbf{l}$ not in $\mathcal{M}$, braids non-trivially with at least one quasi-particle in $\mathcal{M}$: there exists $\mathbf{m} \in \mathcal{M}$, $\frac{\theta_{\mathbf{ml}}}{2\pi} = \mathbf{m}^T K^{-1} \mathbf{l} \notin \mathbb{Z}$.

When the couplings $U_m$ are large, the quasiparticles in $\mathcal{M}$ condense, and the boundary is gapped. Take the $\mathbb{Z}_2$ topological order with $K_{\mathbb{Z}_2} = \begin{pmatrix} 0 & 2 \\ 2 & 0 \end{pmatrix}$ as an example. There are two types of Lagrangian groups: $\mathcal{M}_e = \{(1,0)^T\}$ and $\mathcal{M}_m = \{(0,1)^T\}$. Another example is that in the twisted $\mathbb{Z}_2$ topological order with $K_{\text{twisted } \mathbb{Z}_2} = \begin{pmatrix} -2 & 2 \\ 2 & 0 \end{pmatrix}$, there is only one Lagrangian group, $\mathcal{M}_e = \{(1,0)^T\}$. When the bosons in $\mathcal{M}$ are condensed, the boundary is gapped.

More realistically, we only consider the backscattering of local excitations. Local excitations are created by $e^{i\mathbf{m}_I \phi_I}$, where $\mathbf{m} = K\mathbf{n}$ and $\mathbf{n}$ is an integral vector.

*Example 1. the $\mathbb{Z}_2$ topological order*

Let us take the $\mathbb{Z}_2$ topological order as an example. Including backscattering terms of local excitations that break as most symmetry as possible,

$$\delta S = -\int dt\, dx\, (u_1 \cos 2\phi_1 + u_2 \cos 2\phi_2)\,. \tag{A.5}$$

In the limit $|u_1| \gg |u_2|$,

$$\langle e^{i\phi_1} \rangle \neq 0\,. \tag{A.6}$$

The anyon in the Lagrangian group $\mathcal{M}_e$ is condensed.

And in the other limit, $|u_1| \ll |u_2|$,

$$\langle e^{i\phi_2} \rangle \neq 0\,. \tag{A.7}$$

The anyon in the Lagrangian group $\mathcal{M}_m$ is condensed.

When the electric charge is condensed, $\langle e^{i\phi_1} \rangle \neq 0$, we have an open quasi-string excitation with end points on the boundary, $e^{i\phi_1(x_1) - \int_{\mathcal{L}_{12}} a_1 - i\phi_1(x_2)}$, where $\mathcal{L}_{12}$ is a path from point $x_1$ to point $x_2$. The gapped boundary is now described by a topological action. To derive it, we use the equation of motion in $S^{\text{bulk}}$: $da_1 = 0$, whose solution is $a_1 = d\lambda$ and $\lambda|_{\text{bdry}} = \phi_1$,

$$S_e^{\text{bdry, top}} = \frac{2}{4\pi} \int_{\partial M} \phi_1 da_2\,, \tag{A.8}$$

where $\phi_1 \in [0, 2\pi)$ is pinned to a value $-i\log\langle e^{i\phi_1} \rangle$. Excitations are created on the end points of $e^{i\int a_2}$. Since such end points cost energy, its dynamics is not captured by the topological action.

Similarly, when the magnetic charge is condensed, the gapped boundary is described by

$$S_m^{\text{bdry, top}} = \frac{2}{4\pi} \int_{\partial M} \phi_2 da_1\,. \tag{A.9}$$

*Example 2. Twisted $\mathbb{Z}_2$ topological order.*

In this case,

$$S^{\text{bdry}} = S_0[K_{\text{twisted } \mathbb{Z}_2}] + \delta S\,,$$

$$\delta S = -\int dt\, dx\, (u_1 \cos 2\phi_1 + u_2 \cos 2\phi_2)\,. \tag{A.10}$$

Note that only anyons in the Lagrangian group can be condensed on the boundary, hence the only compatible condensation is described by

$$\langle e^{i\phi_1} \rangle \neq 0\,. \tag{A.11}$$

The other quasi-particle associated with $e^{i\phi_2}$ cannot condense, since the it is not bosonic, and not part of any Lagrangian group.

# B Derivation of the unitary operator that swaps boundaries

The unitary operator (20) comes from gauging the symmetric finite depth circuit that creates a two-dimensional $\mathbb{Z}_2$ SPT on the boundary $\partial\mathcal{V}$ of a volume $\mathcal{V}$, in a three-dimensional $\mathbb{Z}_2$ paramagnetic state [36]. On a three-dimensional triangulated lattice with one qubit on each site, the symmetric circuit is

$$
\begin{aligned}
U_{SPT}(\mathcal{V}) &= \prod_{\triangle_{0123}\in\mathcal{V}} \nu_3(g_0, g_1, g_2, g_3) \\
&= \prod_{\triangle_{0123}\in\mathcal{V}} (-1)^{g_0^+ g_1^- g_2^+ g_3^- + g_0^- g_1^+ g_2^- g_3^+}, \\
g_i^{\pm} &= \frac{1 \pm Z_i}{2},
\end{aligned}
\tag{B.1}
$$

where $\nu_3 \in H^3[\mathbb{Z}_2, U(1)]$ is the 3-cocycle satisfying $\nu_3(g_a, g_b, g_c, g_d) = \nu_3(1, g_a^{-1}g_b, g_a^{-1}g_c, g_a^{-1}g_d)$. It is related to another convention by $\omega(g_a, g_b, g_c) = \nu_3(1, g_a, g_a g_b, g_a g_b g_c)$. In our choice, the non-trivial elements of the cocycle are $\nu_3(1, g, 1, g) = \nu_3(g, 1, g, 1) = -1$, where $g$ is the generator of $\mathbb{Z}_2$ group.

The unitary operator has two further properties:

(1) The unitary operator commutes with the $\mathbb{Z}_2$ symmetry of the $\mathbb{Z}_2$ SPT.

(2) In a closed system, the unitary equals to the identity.

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
