# Peer review of "Boundary states of Three Dimensional Topological Order and the Deconfined Quantum Critical Point"

_SciPost Physics, doi:SciPost Phys. 15, 231 (2023)_

## Round 1 · Referee Report · Anonymous (Referee 1) · 2023-7-5

Strengths

  1. The authors give a very nice exploration on the three elementary gapped boundaries of the 3D $Z_2$ toric code from various perspectives: gauging, layered construction, solvable lattice model, and field theory.

  2. They also construct other gapped boundaries, which host anyons that are confined to the boundary.

  3. An emergent $Z_2^d$ unitary, corresponding to sweeping a 2D $Z_2$ SPT through the bulk before gauging, is carefully studied. The authors write down an explicit operator for the unitary, and construct a gapped boundary that is symmetric under $Z_2^d$.

  4. Existence of multi-critical point in the boundary phase diagram and its connection to DQCP are discussed.

  5. The paper is clean and easy to read.

Weaknesses

No major weakness spotted.

Report

It is a nice work on gapped boundaries of 3D topological orders. The 3D $Z_2$ toric code is among the simplest examples of three 3D topological order, so this study is illuminating for general studies in the future. I recommend it for publication in SciPost.

Requested changes

  1. Are the signs in Eq. (33) correct? In the current form, if we take $\alpha=1$ and $\lambda=0$, then $H = -H^m$, whose boundary is not the "twisted'' boundary as shown in Fig. 8.

  • validity: top
  • significance: high
  • originality: high
  • clarity: top
  • formatting: excellent
  • grammar: excellent

Author:  Wenjie Ji  on 2023-10-24  [id 4060]

(in reply to Report 1 on 2023-07-05)

Thank you for the positive comments, and for pointing out the typo in Eq. (33). We have corrected it.

---

## Round 1 · Referee Report · Anonymous (Referee 2) · 2023-7-8

Strengths

1-The topic, of gapped boundary types of 3+1d topological orders and their phase transition, is relatively new and not very well understood in general.

2-The proposed deconfined quantum critical point (DQCP) description of the boundary multi-critical point is new.

3-The focus on $Z_2$ toric code and its three elementary boundaries makes this paper (potentially) readable to readers with mixed backgrounds.

Weaknesses

1-In the paper, a few theoretical approaches are used: lattice model, layer construction, and effective (quantum) field theory. While different perspectives can add up (which is a good thing), this can require nontrivial effort to make the logic clear. For instance, why for some discussions, a particular approach is discussed, not others. I get the feeling that the authors can do a better job explaining these choices with slight effort.

2-Quite a few results are mentioned along with the leading results. My understanding is that the DQCP conjecture is the main thing. The three boundary types constructed here are studied (or expected) in previous literature. However, I am not sure if this is what the author had in mind because I do not see the main result getting more detailed support compared with other things. This is another potential thing to improve.

Report

It took longer than I thought to finish reviewing this manuscript. This manuscript describes three “elementary boundaries” of 3d $Z_2$ toric code. It also conjectures that the (possibly exist) multi-critical point at the junction of three boundary phases can be described by a deconfined quantum critical point (DQCP). It is argued that an emergent $\mathbb{Z}_2^d$ becomes part of the global symmetry of DQCP.

From my reading of the manuscript and a few related references, I believe that most of the results stated in the draft are reasonable. I believe they are likely to be correct. I also think the conjecture about DQCP is valid in the sense it sits at the edge of our knowledge. I think this paper deserves consideration for publication as a research paper.

On the other hand, I think there are some possible improvements the author may like to consider. After the authors reply to my questions and consider a few suggested improvements, a recommendation for publication is promising.

1-The emergent symmetry in Eq.(20) preserves the ground subspace of the bulk, and it changes the "deconfined" and the "twisted" boundary. The author commented that the unitary operator $U(\mathcal{V})$ is not onsite. This operator $U(\mathcal{V})$ seems to be central to several discussions, and therefore I want to ask a few very specific questions:

(1a) The authors wrote the explicit expression of $U(\mathcal{V})$ in the lattice model and commented that it is not onsite. (It is emergent.) How about the circuit depth?

I ask because the authors argued in a later section that the symmetry operator changes the statistics of the endpoints of the m-loop. So, I suppose $U(\mathcal{V})$ cannot be a finite depth circuit? Is that right?

(1b) I realize that the discussion of $U(\mathcal{V})$ is mainly in the lattice model approach. Later, the discussion of the self-statistics of $m$-loop is mainly in the layer construction approach. Is it that each approach can solve the problem, but the authors wanted a particular simple picture, and that was the reason to switch between different approaches? If this is the case, some explanation of this point can be beneficial.

(1c) I believe the statistics of end-points of the $m$-loop discussed in section 3.1 is reasonable to some extent; reading section 3.1 as well as Ref. [22]. They helped me gain some intuition.

On the other hand, I have a general question. In general, how do people define the statistics of $m$-loop that ends at a boundary surface (as shown in FIg. 5)? (It is hard for me to translate the discussions of [22] to the end-point statistics. I also do not immediately see how the definition of $S$-matrix and $T$ in any context generalizes to this case.) If the approach discussed in section 3.1 is original, and not appeared anywhere, perhaps it deserves a more detailed explanation.

(1d) Is there a field theory description of the operator $U(\mathcal{V})$ discussed (either here or in previous literature)?

2- About the phase transition:

(2a) For the boundary Hamiltonian Eq.(33), out of curiosity, where the point $\lambda=0$, $\alpha=1/2$ lie in the phase diagram (Fig. 8)?

(2b) I was wondering if the last few sentences of the bottom paragraph on Page 14 (start with "A similar phase diagram was studied numerically in Ref. [48] ...") were a partial justification of the phase diagram. Do I understand correctly that the authors can map a particular form of Eq.(33) to the model studied in [48]. Thus, we can justify the conjecture of this manuscript by the numerical in [48]? If so, this sounds like valuable support, and I suggest the author make the statement more pronounced.

3- About the emergent $\mathbb{Z}_2^d$ symmetry.

(3a) The notation $\mathbb{Z}_2^d$ appeared in section 3.2. If I read the draft correctly, this is the first time the notation appears and is explained. However, I cannot identify the explanation of the upper index $d$. Can the author explain what $d$ means? (An even earlier place $\mathbb{Z}_2^d$ appeared was in the introduction and the abstract. The notation was even more confusing in those early places (e.g. I thought it was $\mathbb{Z}_2$ to some power $d$, which is surely not what the authors mean.) Some clarification of notation should be useful.

(3b) Related to the discussion in section 4.2. Likely this is not what the authors thought, but is there a situation that the $\mathbb{Z}_2^d$ can be treated as an onsite symmetry at the multi-critical point? Here are two conflicting thoughts I can think of. One argument against onsite is that even at the critical point, the unitary acts in the bulk in a nontrivial way. On the other hand, maybe the operator acting on the ground state is equivalent to something simpler acting on the ground state. In other words, can the operator be equivalent to an onsite operator when it acts on the ground state?

Requested changes

1-improve the discussion of notation $\mathbb{Z}_2^d$, e.g., explain it when it first appear.

  • validity: good
  • significance: high
  • originality: high
  • clarity: good
  • formatting: excellent
  • grammar: excellent

Author:  Wenjie Ji  on 2023-10-24  [id 4059]

(in reply to Report 2 on 2023-07-08)

(1a) Reply: The operator is a depth-1 quantum circuit, with support on a 3-dimensional volume. And it changes the statistics of the endpoints of $m$-loop on the 2-dimensional boundary. If the unitary were supported on a 2-dimensional surface, it would not be of finite depth, as the referee anticipated.

(1b) Reply: Thank you for reminding us the switch of approaches in our presentation. We have added an explaination making connections between the two approaches.

(1c) Reply: In general, the statistics can be defined using topological field theory as the contribution in the topological partition function. In particular, one can think of a space-time process that flips the entire string such that the two end points (located on the boundary) are exchanged. The correlation function of this process gives a phase factor, which can be used to define the self statistics of the end point. Nevertheless, to translate this definition to one that is rigorous at the lattice level is an interesting open problem.

(1d) Reply: We have provided a field theory description of this defect in Subsection III.C.

(2a) Reply: For the specific lattice Hamiltonian we have chosen, $\lambda=0, \alpha=1/2$ realizes the ferromagnetic phase. In general, the fate of the point can depend greatly on the form of $H_{SPT}$. See Refs. 45-48 which finds various intermediate phases depending on the choice of (or sign of) $H_{SPT}$.

(2b) Reply: Along the $\lambda=0$ line, the boundary Hamiltonian can be mapped exactly to the Hamiltonian studied in [48]. There, the Hamiltonian has $\mathbb{Z}_2^3$ symmetry and here we simply can restrict to the diagonal $\mathbb{Z}_2$ symmetry. The difference is the $\lambda$ term here is nearest neighbor Ising interaction which only preserves the $\mathbb{Z}_2$ symmetry, while the $\lambda$ term in [48] is a next-nearest neighbor Ising interaction which preserves the full $\mathbb{Z}_2^3$ symmetry. Thus the resulting phase diagram for $\lambda \ne 0$ can be different.

(3a) Reply: The $d$ here refers to the ``duality" transformation which swaps two of the boundaries. It can be also thought of as corresponding to sweeping an invertible ``defect" present in the $\mathbb{Z}_2$ toric code. The intention of the superscript is to avoid confusion with the $\mathbb{Z}_2$ symmetry which gauges to the toric code. To avoid confusion, we have revised the notation to $\mathbb{Z}_{2,\text{d}}$.

(3b) Reply: It is possible to make the $\mathbb{Z}_{2,\text{d}}$ symmetry onsite in the ground state subspace. An example to realize this is to start from a $\mathbb{Z}_2 \times \mathbb{Z}_{2,\text{d}}$ SPT and gauge the $\mathbb{Z}_2$ symmetry. The resulting model will be a $\mathbb{Z}_{2,\text{d}}$ symmetry-enriched toric code with $\mathbb{Z}_{2,\text{d}}$ acting onsite in the ground state subspace.

Anonymous on 2023-11-08  [id 4099]

(in reply to Wenjie Ji on 2023-10-24 [id 4059])

The authors have provided satisfactory answers to my questions. I recommend the publication.

---

## Round 2 · Referee Report · Anonymous (Referee 2) · 2023-11-8

Report

The authors have provided satisfactory answers to my questions. I recommend the publication.

---

## Editorial Decision

published